# Identifying Lymph Nodes and Their Statuses from Pretreatment Computer Tomography Images of Patients with Head and Neck Cancer Using a Clinical-Data-Driven Deep Learning Algorithm

**DOI:** 10.3390/cancers15245890

**Published:** 2023-12-18

**Authors:** Sheng-Yao Huang, Wen-Lin Hsu, Dai-Wei Liu, Edzer L. Wu, Yu-Shao Peng, Zhe-Ting Liao, Ren-Jun Hsu

**Affiliations:** 1Institute of Medical Science, Tzu Chi University, Hualien 970374, Taiwan; martin12345m@gmail.com (S.-Y.H.); dwliu5177@yahoo.com.tw (D.-W.L.); 2Department of Radiation Oncology, Hualien Tzu Chi General Hospital, Buddhist Tzu Chi Medical Foundation, Hualien 970473, Taiwan; wlhsu@tzuchi.com.tw; 3Cancer Center, Hualien Tzu Chi Hospital, Buddhist Tzu Chi Medical Foundation, Hualien 970473, Taiwan; 4School of Medicine, Tzu Chi University, Hualien 970374, Taiwan; 5DeepQ Technology Corp, New Taipei City 242062, Taiwan; edzer_wu@deepq.com (E.L.W.); ys_peng@htc.com (Y.-S.P.); zt_liao@htc.com (Z.-T.L.)

**Keywords:** head and neck cancer, computed tomography, deep learning, semantic segmentation, image processing

## Abstract

**Simple Summary:**

We proposed a deep learning algorithm to detect lymph nodes and classify them in the head and neck region on computed tomography. We further analyzed the inference result from the model and found that the size of the lymph nodes may be a characteristic for the model to classify them. This finding is consistent with current clinical aspects. We will deploy the model in clinical practice and hope to assist clinicians in finding out the lesions more correctly and efficiently.

**Abstract:**

Background: Head and neck cancer is highly prevalent in Taiwan. Its treatment mainly relies on clinical staging, usually diagnosed from images. A major part of the diagnosis is whether lymph nodes are involved in the tumor. We present an algorithm for analyzing clinical images that integrates a deep learning model with image processing and attempt to analyze the features it uses to classify lymph nodes. Methods: We retrospectively collected pretreatment computed tomography images and surgery pathological reports for 271 patients diagnosed with, and subsequently treated for, naïve oral cavity, oropharynx, hypopharynx, and larynx cancer between 2008 and 2018. We chose a 3D UNet model trained for semantic segmentation, which was evaluated for inference in a test dataset of 29 patients. Results: We annotated 2527 lymph nodes. The detection rate of all lymph nodes was 80%, and Dice score was 0.71. The model has a better detection rate at larger lymph nodes. For those identified lymph nodes, we found a trend where the shorter the short axis, the more negative the lymph nodes. This is consistent with clinical observations. Conclusions: The model showed a convincible lymph node detection on clinical images. We will evaluate and further improve the model in collaboration with clinical physicians.

## 1. Introduction

Head and neck cancers have remained among the ten leading causes of cancer-related death in Taiwan for a long time [1]. They include oral cavity, oropharynx, hypopharynx, larynx, and nasopharynx cancers. Most head and neck cancers are associated with life habits, such as smoking, drinking alcohol, and chewing betel nuts.

The head and neck region has abundant lymphatic drainage [2]. The status of lymph node metastasis is critical to prognosis [3], including the location and number of cancer-involved lymph nodes and presentation of extranodal extension (ENE) [4].

Most head and neck cancers are treated surgically if eradicable, even at an advanced stage [5]. Tumors with involved lymph nodes that are completely removed have better prognoses than others [6]. Typically, surgeons conduct a clinical workup before operating to gather additional information for selecting appropriate surgical techniques, which may involve dissecting the cervical lymph nodes. The workup also affects the treatment choice in neoadjuvant, adjuvant, or even definitive treatment settings.

Medical imaging plays a crucial role in the clinical workup, with various tools available, including computer tomography (CT), magnetic resonance imaging (MRI), and positron emission tomography (PET) [5,7]. Each imaging tool offers unique advantages. For example, CT effectively finds bony invasion, while MRI excels at delineating soft tissue involvement. In contrast, PET can provide a comprehensive diagnosis of locoregional and distant metastasis by measuring cell activity using F-18 [8]. The efficacy of using medical images as clinical diagnostic tools has been evaluated [4,9]. Some features of lymph nodes related to morphology or enhancement on images may indicate tumor involvement: a shorter axis of a lymph node of >1 cm, heterogeneous enhancement, or rough border of a lymph node, which might be a sign of ENE [10]. However, even when interpreted by well-experienced clinical physicians or radiologists, the sensitivity and specificity of CT images were 72% and 83%, respectively, while the area under the receiver operating characteristic curve (AUC) was 0.65–0.69 [11,12]. In contrast, the sensitivity and specificity with MRI were 0.7–0.8 and 0.5–0.7, respectively [13,14].

Efficient and correct identification and delineation of lymph nodes is crucial for clinical diagnosis, surgical techniques, and other treatments. Traditionally, radiologists or clinical physicians such as otolaryngologists would have to view CT images to obtain information on clinical diagnosis and make treatment decisions. In Taiwan, it is common for an experienced otolaryngologist to have more than a hundred patients in one outpatient clinic, and serve more than twenty in inpatient at the same time. Even with the resident’s assistance, this is still overwhelming. Although not all patients are diagnosed with head and neck diseases, it creates time pressure for physicians to read images, make decisions, and discuss with patients. In Hualien Tzu Chi Hospital, head and neck CT examinations are generally carried out within a week; most of them should be reported and submitted within 2 weeks, which is also exhausting for the staff. For junior residents, it would take more time and effort to complete image reading. An automated assistant for clinical diagnosis might relieve the loading.

There has been a recent trend toward using digital systems to assist clinical diagnosis. These systems analyze data from laboratory tests, medical records, and images to generate results for clinical needs, such as establishing clinical impressions, alerting for emergencies, or risk stratification. Among these digital systems, deep learning-based computer vision techniques have made significant progress in analyzing medical images [15].

Convolutional neural networks (CNNs) have been widely used in deep learning for computer vision tasks, including classification, object detection, and semantic segmentation [16]. Models derived from ResNet [17] or VGG [18] were used for classifying regions of interest by human experts. Fully convolutional networks like Unet [19], on the other hand, can segment the targets from medical images. Such models have been used to study lymph node status, or segmentation tasks at head and neck region. Using a CNN model, Kann et al. classified lymph nodes segmented by experts as normal or tumor-involved in CT images, achieving an impressive AUC of 0.91 [20]. The model was composed of a 3D model and a size-invariant model and was able to extract features while preventing itself from overfitting [20]. Another study examined segmentation for head and neck lymphatic drainage areas [21], which can be applied to contouring in radiotherapy. In this study, a fully convolutional neural network was proposed to deal with segmentation for head and neck lymphatic drainage area.

However, lymph nodes’ inconsistent morphology and size make determining their status and delineation challenging. Lymph nodes can range in diameter from being almost invisible in medical images to >10 cm. Moreover, their 2D projections can appear with similar textures to other structures in image slices, such as vessels, muscles, or salivary glands. Another challenge in this task is annotation, a time-consuming process for clinical physicians to segment and label the lymph nodes for classification.

The most challenging issue is deploying a model in the clinical field. While having sufficient data can increase the likelihood of constructing a well-performing model, additional factors must also be considered for successful deployment. A performance gap between training and real-world data has been reported [22], and factors such as the examination settings, presentation of inference results, and the specific needs of clinicians and other healthcare professionals can all affect the model’s effectiveness.

To address these challenges, we present a novel approach combining deep learning models, image processing algorithms, and domain knowledge segment and classify lymph nodes. The proposed method is further evaluated based on clinical knowledge to assess the reliability of the inference results.

## 2. Materials and Methods

### 2.1. Study Cohort

We retrospectively enrolled patients diagnosed with oral cavity, oropharynx, hypopharynx, or larynx cancers at Hualien Tzu Chi General Hospital between 1 January 2008, and 31 December 2018. These patients had confirmed diagnoses from biopsies carried out at our hospital and should receive surgery as definitive treatment. We collected pre-surgery contrast-enhanced head and neck CT images and surgical pathology reports. We collected pretreatment CT images if a patient received definitive concurrent chemoradiotherapy without surgery. Patients diagnosed or treated due to other cancers before would be excluded.

Appendix A shows the patients enrolled in this study. After excluding CT images with low resolutions or poorly identified targets, the final dataset included 271 patients with 274 CT image series (Figure 1). These patients were randomly divided into training (*n* = 213), validation (*n* = 30), and testing (*n* = 28) sets. The numbers of patients and lymph nodes are reported in Appendix A.

### 2.2. Image Prepare and Annotations

Two clinical physicians and a radiologist reviewed the CT images, after which the clinical physicians segmented and classified the lymph nodes on the CT images. The radiologist provided advice to the physicians in case of any uncertainty. Two pathologists reviewed the pathology reports, and annotations were created to classify lymph node status based on the pathology report, which will serve as the ground truth. We used the DeepQ AI platform (https://www.deepq.ai/?lang=en, accessed on 20 September 2023) (from DeepQ, New Taipei City, Taiwan) to annotate images, which were deidentified before upload.

### 2.3. Model and Training Methodology

#### 2.3.1. Model

The nn-UNet model integrates most state-of-the-art semantic medical image segmentation techniques [23]. It extracts features from images in different spacing in two stages to prevent the model from losing complete picture information when training in image patches. Computing resources can be preserved by training in patches. The model is trained by self-adjusting hyperparameters based on data features (i.e., sample size, image size, spacing, and modalities). The framework automatically defines the batch size, number of epochs, model architecture, and learning rate. However, if necessary, the user can manually modify them based on past experiences or competition on different open datasets. We constructed a model based on nnUNet to fit our situations.

First, we preprocessed images. We set pixel spacing as 0.89 × 0.48 mm according to the value obtained from the dicom file. Windowing and intensity were normalized by window level and width.

We chose a 3D network from clinical aspects. The morphology of lymph nodes may be confused with other structures around them in 2D projections, such as vessels, glands, muscles, or other soft tissues, that become distinct in 3D projections. We expected that the result would be better in a 3D network. The nn-UNet network will automatically adjust its architecture to handle spacing anisotropy between axes. Specifically, the network applies convolution and pooling operations to high-resolution axes until the resolution factor between axes becomes <2. This approach ensures that the model extract feature is unaffected by the varying resolution between axes. We used a patch size in the network of 160 × 192 in the first stage and 192 × 224 in the second stage. We inherited the loss function in nnUNet, which combines Dice loss and cross-entropy. Finally, post-processing was carried out to filter prediction masks with pixel numbers below the threshold. The details for model settings were summarized in Appendix A.

We evaluated the model’s inference results using two metrics: Dice score and detection rate. The Dice score (*s*) was calculated as follows:(1)$$s=\frac{2\left|P+GT\right|}{\left|P\right|+\left|GT\right|}$$
where *P* represents prediction, and *GT* represents ground truth (label). The detection rate (*d*) was calculated as follows:(2)$$d=\frac{TP}{TP+FP}$$
where *TP* represents true positive, and *FP* represents false positive. Both evaluation metrics were calculated by each image slice and averaged to represent each study.

#### 2.3.2. Training Method

We trained the model for 300 epochs with a mini-batch size of 2. We used stochastic gradient descent as optimization with an initial learning rate of 0.01 and gradually decaying the learning rate as training progressed. An oversample technique was used to address class imbalances. Specifically, we ensured that >33.3% of the patches contained at least one positive mask. In addition to using the default data augmentation provided by nnU-Net [23], we used random translation to ensure the mask was distributed uniformly within the patches, which can improve model robustness and generalization. All experiments were trained on a single GeForce GTX 1080 Ti graphics processing unit. The training, validation, and inference were performed with Pytorch (version 1.11.0) in Python 3.9.

## 3. Results

### 3.1. Basic Image Features

This study included 2527 lymph nodes annotated from 271 patients. Appendix A shows that most lymph nodes were annotated as negative.

#### Distribution of Lymph Node Size and Intensity

We analyzed the lymph nodes according to their short axis size (Table 1a), finding a trend where the shorter the short axis, the more negative the lymph node. The pixel intensity in lymph node regions showed greater diversity in negative than in other lymph nodes.

### 3.2. D Model Performance

#### 3.2.1. Performance Evaluation

First, we examined the relationship between the threshold of the portion overlapped and the detection rate (Appendix A). We found that the detection rate reached >60% at a threshold of 50%. We then evaluated the model using two settings: >0% and >50%.

We trained models with different manipulations: combine all three annotation classes, combine only the positive and ENE classes, and separate classes. First, we evaluated the ability to detect lymph nodes in three models (Appendix A). The three models showed consistent Dice scores and detection rates. Then, we evaluated the inference of the model trained on separate classes (Appendix A). The detection rates for negative, positive, and positive with ENE lymph nodes were 76%, 73%, and 90%, respectively. The average lymph node detection rate was 80%, while the Dice score was 0.71.

#### 3.2.2. Inference Analysis

##### The Model Can Size Classify Lymph Nodes

We analyzed lymph nodes detected by the model in the test set to determine the clinical characteristics it may capture. Table 1b shows that, of the 176 identified negative lymph nodes, 167 had a short axis <1 cm. In contrast, all identified positive lymph nodes with ENE had a short axis >1 cm. Figure 2 shows some of the model’s accurate predictions.

##### False Negative/Positive Inferences

We found that Dice scores decreased when inferring separated classes, especially for the positive and ENE classes. We analyzed accuracy at the pixel level to determine possible reasons for this. Appendix A shows the confusion matrix. Most of the misclassified pixels were recognized as background (Appendix A). The model also predicted some background pixels as lymph nodes (Appendix A).

We also found a correlation between the detection rate and lymph node size (Appendix A). The detection rate was 45% for lymph nodes with a short axis <5 mm. The detection rate increased to >80% for larger lymph nodes.

##### Misclassification of Positive and ENE Lymph Nodes

Appendix Ac shows that the model classified 11% of pixels labeled as positive lymph nodes as ENE and 11% labeled as ENE as positive lymph nodes. We evaluated the performance of models on the test set with combined P and ENE classes to determine how misclassification affected Dice scores (Table 2). The models trained on all separate classes or combined P and ENE classes showed better Dice scores and detection rates on the test set with combined classes than on that with P and ENE classes separated.

## 4. Discussion

### 4.1. Applying a Deep Learning Model in Classifying Lymph Mode Metastasis in Head and Neck Cancer

There is limited research on applying machine learning or deep learning algorithms to lymph nodes in patients with head and neck cancers. One reason for this is data collection. There are open datasets for medical imaging, including contouring and classification of organs and lesions [16], but almost none for head and neck cancers. A dataset is the basis for training and evaluating a model, and establishing such a novel dataset would be time- and labor-intensive, especially for clinical practitioners. Our study enrolled more patients than previous studies [20,21], and the annotated lymph nodes are also comparable. To our knowledge, this is the first study to identify lymph nodes and statuses using semantic segmentation. We hope that a model trained using such a volume of data could be helpful clinically.

### 4.2. Model Inference

#### 4.2.1. Detection Rate and Dice Score

This study aimed to assist clinicians in detecting lymph nodes from medical images to make diagnoses and decide on treatments, but not to replace clinicians or screen through serial images. Clinical physicians should still review the images to check the model’s suggestions and the primary tumor’s extension. It marks potentially involved lymph nodes and reminds physicians while reviewing image slices. Therefore, we are more concerned with classification accuracy than detailed object contouring. However, since classifying an entire image based on the presence of a tumor-involved lymph node is meaningless in clinical practice, object detection remains necessary. We assigned the task to semantic segmentation due to its visual result presentation. A questionnaire survey in our hospital showed that 82% of physicians preferred lymph nodes to be presented as segmentation masks rather than bounding boxes, mainly due to visualization. When clustered or serial lymph nodes are present, segmentation masks could be easier to read than stacked bounding boxes.

Evaluating lymph node segmentation is challenging. While tumor-involved lymph nodes can be >5–6 cm, most objects are <1 cm (Table 1a). In addition, the anatomy and structure are complicated in head and neck lymphatic drainage regions. They contain many vessels and glands whose size, texture, and even intensity are similar to lymph nodes, which can interfere with the model recognizing lymph nodes.

Our model had higher detection rates than Dice scores. These metrics are quite different: the Dice score considers false negatives and positives, while the detection rate considers whether the model “captures” the object, meaning its marking of pixels labeled as ground truth. The detection rate calculation may underestimate the false positive effect. We evaluated the false positive rate (Appendix A), finding 1–4 false positive components per case, depending on the class. This false positive rate should be tolerable for clinical practice, although further evaluation after deployment is necessary.

Some studies have evaluated model performance in identifying lymph nodes from medical images by detection rate [24,25], reporting a detection rate of 0.7–0.8 and a false positive rate of 10 per volume. Our study showed a better overall detection rate of 0.8 and a false positive rate of 2.36 per volume. Among classes, the best detection rate was for ENE (0.9). Among sizes, there was a better detection rate for lymph nodes with a short axis >5 mm (>0.7; Appendix A), which was also better than in a previous study (0.62). Our improved results may be due to nnUNet’s comprehensive feature extraction, especially for spatial information.

We examined the relationship between the threshold and detection rate, which is the fraction of the ground truth mask overlapped by the inference mask (i.e., true positive; Appendix A). We obtained a detection rate of >0.6 even at a threshold of >0.5. The model could determine the location and size of those detected lymph nodes. We believe our model will be sufficiently robust as an alarm system in clinical practice.

#### 4.2.2. Effects of Clinical Features on Inference

The model might classify lymph nodes according to their size. Table 1b shows that smaller lymph nodes tended to be classified as negative, and those with a short axis >1 cm were more likely to be tumor-involved. The short axis of all identified lymph nodes with ENE was >1 cm, consistent with current clinical experience that one feature indicating malignant lymph node changes is size, usually defined as a short axis >1 cm. It can be referred to as an interpretability of the model from a clinical point of view, and convinces the clinical physicians when the model alarms at a specific lymph node during practice.

#### 4.2.3. The Potential of a Model Trained on Images Generated Using Different Protocols at Different Timepoints

We retrospectively collected images over 10 years. During these years, computed tomography machines, settings for examination, and protocols have changed several times. Even with image preprocesses such as intensity normalization and clipping, intensity enhancement, and contrast remained confusing (Appendix A). The model could obtain convincing inferences on test datasets. The images might be heterogeneous in real-world data from different examination machines and protocols. It is common to see gaps in model performance between model training datasets and deployment [22,26]. One reason for this gap is the variance between training and real-world data. Several studies have examined approaches to address this problem, such as transfer learning [27] or domain adaptations [28]. However, labeled data are still necessary, and the effect is not always promising. The model trained on a heterogenous dataset may have better adaptability after deployment but not reach the perfect performance shown in their original studies.

#### 4.2.4. P and ENE Class Misclassification

We found the model confused pixels labeled positive and positive with ENE lymph nodes. Since both classes are tumor-involved, they may share some common features from a clinical perspective, such as larger size and central necrosis. The status of these two classes makes a difference in staging and prognosis but not treatment choice. Since they are clinically suspected of malignancy, dissection during surgery or dose escalation during radiotherapy will be preferred. Therefore, it can still be valuable to classify lymph nodes as tumor-involved or not as an alarm system for clinical practice.

### 4.3. Limitations

#### 4.3.1. No Consistent Image Examination Protocol

A consistent image examination protocol is still necessary to improve accuracy. Those protocols are established in clinical practice according to modalities, examination aims, targets, and clinical needs. The aim and target were specific in our case, but image quality varied over time. Further evaluation and model adjustment showed that brightness and contrast should be clipped in a range to maintain consistent intensity for corresponding structures in the images, which may lead to stable inference results.

#### 4.3.2. Improved Classification Ability for P and ENE Classes

Future work will aim to reinforce the model’s ability to classify P and ENE lymph nodes, potentially by increasing the number of ENE annotations since they were much fewer than for the other two classes (Appendix A).

## 5. Conclusions

We present a model trained with semantic segmentation to identify lymph nodes and their tumor-involved statuses. Our model had a satisfactory detection rate, but its Dice score could be improved. After deployment, we will evaluate and further improve our model in collaboration with clinical physicians, including model adjustments and image examination protocols. Figure 3 shows our expectations for our model’s clinical contribution after deployment.

In the future, we would like to explore more about the effect of data heterogenicity on model performance. The size of the lymph nodes will be recorded to confirm if the trend that smaller lymph nodes tend to be classified as negative is consistent. We will establish a protocol for CT examinations to obtain stable images, and analysis about intensity or other radiomics features could be applied. We hope that the result of further research could make the model more useful in clinical practice, and most importantly convincible.

## Figures and Tables

**Figure 1 cancers-15-05890-f001:**
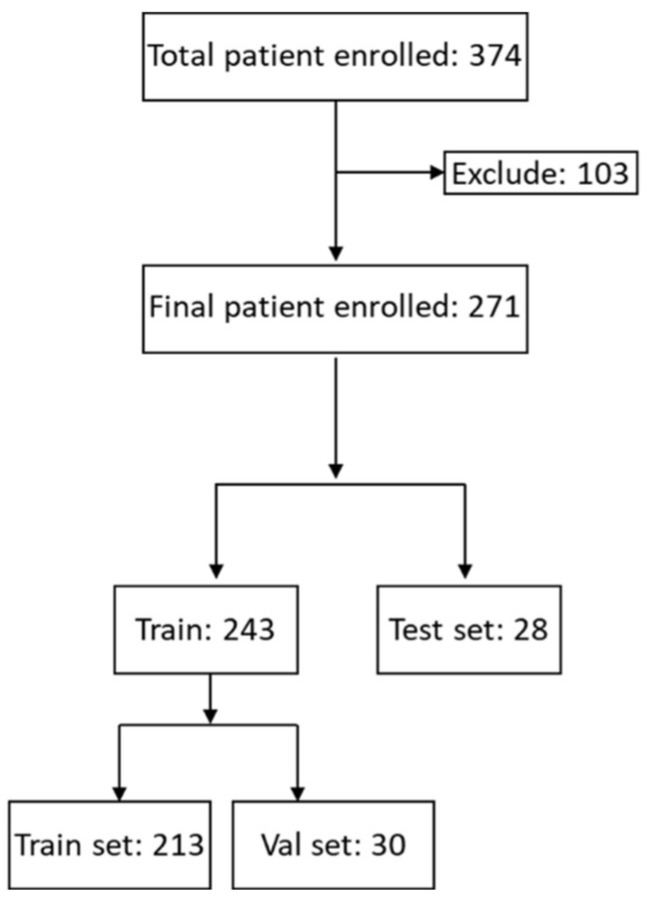
Flowchart of study enrollment. We initially enrolled 374 patients. One hundred and three cases were excluded due to poor resolution of images. Finally, images from 271 patients were included, with 243 in the train set and 28 in the test set.

**Figure 2 cancers-15-05890-f002:**
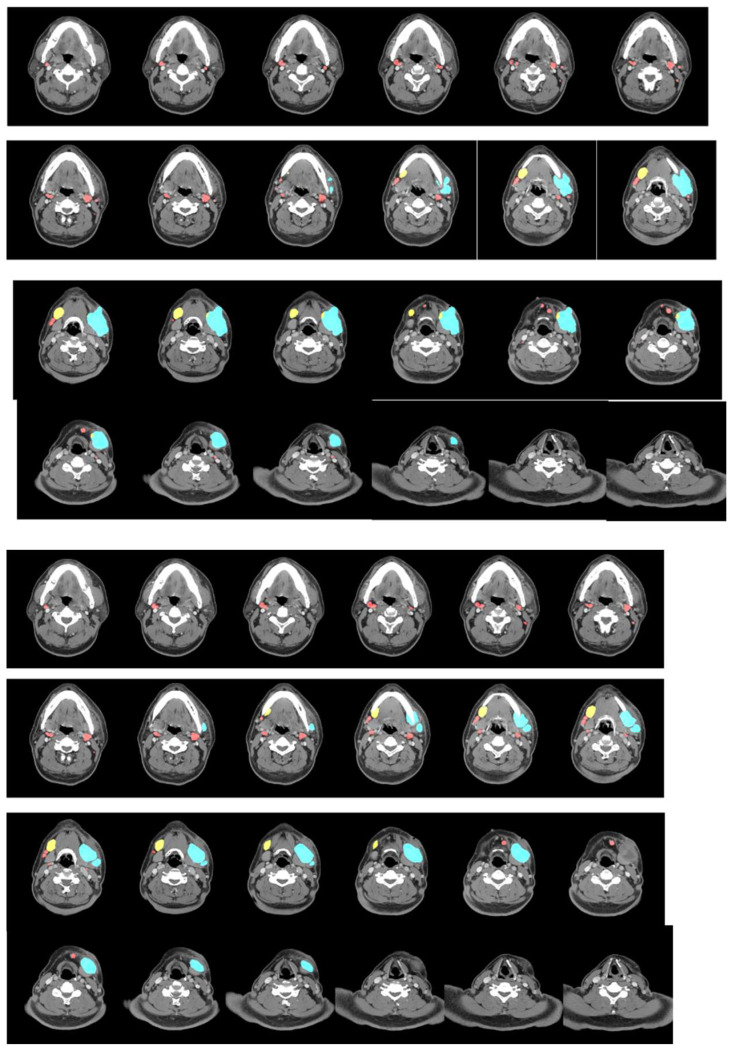
Examples of inference results. (Upper: inference result; middle: ground truth; lower: original image) The inference results compared with ground truth annotation. Red: negative; yellow: positive; light blue: extranodal extension.

**Figure 3 cancers-15-05890-f003:**
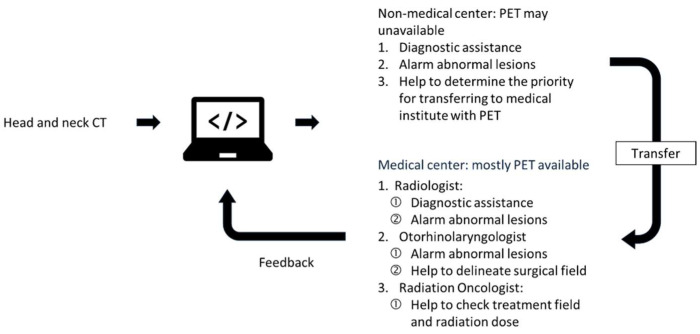
The anticipation of the clinical deployment. We hope the algorithm to fill the gap due to uneven medical examination resources between medical institutes.

**Table 1 cancers-15-05890-t001:** (a)The size (represented by a short axis) of lymph nodes by classes on the dataset. (b) The size of the lymph node by class from the model’s inference result.

(a)									
	Train (%)			Valid (%)			Test (%)		
	<1 cm	>1 cm	Total	<1 cm	>1 cm	Total	<1 cm	>1 cm	Total
N	1470 (96)	61 (4)	1531	270 (96)	11 (4)	281	221 (96)	10 (4)	231
P	125 (45)	153 (55)	278	14 (45)	16 (55)	30	18 (60)	12 (40)	30
ENE	19 (13)	126 (87)	145	2 (13)	14 (87)	16	0	10	10
**(b)**									
**Num (%)**	**<1 cm**			**>1 cm**			**Total**		
N	167 (96)			9 (4)			176		
P	17 (60)			11 (40)			28		
ENE	0			9			9		

**Table 2 cancers-15-05890-t002:** Performance of model on test dataset with P and ENE classes combined. (a) Model trained in separate classes. (b) Model trained in P and ENE classes combined.

3D Metric(%)	(a)		(b)	
	ENE+P	N	ENE+P	N
Detection rate (>0)	87.50	76.58	92.50	75.68
Detection rate (>50)	80.00	59.46	87.50	61.71
FP/image	1.29	4.64	1.11	5.46

## Data Availability

The data were collected from Hualien Tzu Chi General Hospital under the supervision of the Institutional Review Board, and were not allowed to be available to the public.

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
