# Peer review of "Identifying Lymph Nodes and Their Statuses from Pretreatment Computer Tomography Images of Patients with Head and Neck Cancer Using a Clinical-Data-Driven Deep Learning Algorithm"

_cancers, 2023, doi:10.3390/cancers15245890_

Round 1

Reviewer 1 Report

Comments and Suggestions for Authors

- Specify the exact semantic segmentation method used for lymph node detection. Provide a concise overview of the algorithm or architecture, ensuring that readers can understand the technical details.

   - Emphasize the significance of your method in automatically detecting and classifying lymph nodes without the need for manual segmentation. Highlight any advantages or improvements over existing manual segmentation approaches.

   - Elaborate on how the dataset's heterogeneity, stemming from different time periods, examination settings, protocols, and machines, impacts the model's performance. 

   - Provide more evidence or discussion on why the heterogeneity in the dataset might offer better adaptability to the model after deployment.

   - Reinforce the clinical relevance of your findings by further discussing the observed trend towards larger lymph nodes being classified as malignant.

-Consider providing quantitative measures or statistical analyses to support this observation and its clinical implications.

   - Include a brief section on future work, outlining potential avenues for further research or improvements to the proposed method. 

Comments on the Quality of English Language

There are some long lines in the paper 

Author Response

Thank you again for your comments and suggestions.

Reviewer 2 Report

Comments and Suggestions for Authors

hello

thank you very much for an interesting paper

this paper is important to show each clinician the role of new technologies and recent advances in improving overall treatment and diagnostics in OSCC

the abstract is well-written and structured

introduction is ok, please add a referrance to NCCN guidance and the usage of CT/MR in LN and SLN diagnostics

well written introduction

the flow chart is very well ilustrated

how this Dice score is or should be used by other surgeons to evaluate the lymph nodes status?

what software war used to evaluate CT or other radiographs for lymph node evaluation

figure 2 is too big, it should be enlarged in more visible fields, or make not 1 but 6 more improved and more readable CT scans

what were the inclusion and exclusion criteria for the study

what are the study limitations

is it possible to used the following method to asses if a LN is metastatic or just enlarged with some inflammation?

can this CT method also used for MRI? or PET-CT?

is there a possibility to correlate lymph node status withy TNM?

paper is missing limitations

please write 5 key conclusions from this study and highlight them 

Author Response

Thank you again for comments and suggestions.

Round 2

Reviewer 1 Report

Comments and Suggestions for Authors

I have no more suggestions